# The influence of diagnostic subgroups, patient- and hospital characteristics for the validity of cardiovascular diagnoses–Data from a Norwegian hospital trust

**Cathrine Sæthern Rye**[1,2]*, **Anne Pernille Ofstad**[3,4], **Bjørn Olav Åsvold**[5,6], **Pål Richard Romundstad**[7], **Julie Horn**[7,8], **Håvard Dalen**[2,9,10]

1 Department of Medicine, Namsos Hospital, Nord-Trøndelag Hospital Trust, Namsos, Norway, 2 Clinic of Cardiology, St. Olavs University Hospital, Trondheim, Norway, 3 Department of Medical Research, Bærum Hospital, Vestre Viken Hospital Trust, Gjettum, Norway, 4 Medical Department, Boehringer Ingelheim Norway KS, Asker, Norway, 5 Department of Public Health and Nursing, K.G. Jebsen Center for Genetic Epidemiology, Norwegian University of Science and Technology, Trondheim, Norway, 6 Department of Endocrinology, St. Olavs Hospital, Trondheim University Hospital, Trondheim, Norway, 7 Department of Public Health and Nursing, Norwegian University of Science and Technology, Trondheim, Norway, 8 Department of Obstetrics and Gynecology, Levanger Hospital, Nord-Trøndelag Hospital Trust, Levanger, Norway, 9 Department of Circulation and Medical Imaging, Norwegian University of Science and Technology, Trondheim, Norway, 10 Department of Medicine, Levanger Hospital, Nord-Trøndelag Hospital Trust, Levanger, Norway

* Cathrine.Sethern.Rye@helse-nordtrondelag.no

## Abstract

### Background

Cardiovascular discharge diagnoses may serve as endpoints in epidemiological studies if they have a high validity. Aim was to study if diagnoses-specific characteristics like type, sub-categories, and position of cardiovascular diagnoses affected diagnostic accuracy.

### Methods

Patients (n = 7,164) with a discharge diagnosis of acute myocardial infarction, heart failure or cerebrovascular disease were included. Data were presented as positive predictive values (PPV) and sensitivity.

### Results

PPV was high (≥88%) for acute myocardial infarction (n = 2,189) (except for outpatients). For heart failure (n = 4,026) PPV was 67% overall, but higher (>99%) when etiology or echocardiography was included. For hemorrhagic (n = 257) and ischemic (n = 1,034) strokes PPVs were 87% and 80%, respectively, with sensitivity of 79% and 75%. Transient ischemic attacks (n = 926) had PPV 56%, but sensitivity 86%. Primary diagnoses showed higher validity than subsequent diagnoses and inpatient diagnoses were more valid than outpatient diagnoses (except for transient ischemic attack). The diagnoses of acute myocardial

**Data Availability Statement:** We have ensured that all underlying data used in the manuscript will be made freely available from the research

repository of Norwegian University of Science and Technology (DataverseNO) under the CC0 (Creative Commons Zero) license. The dataset named "Replication Data for: The influence of diagnostic >subgroups, patient- and hospital characteristics for the validity of cardiovascular diagnoses – Data from a Norwegian hospital trust." View the dataset at https://dataverse.no/dataset.xhtml?persistentId= doi:10.18710/SNTJJZ The dataset was created in NTNU – Norwegian University of Science and Technology (view at https://dataverse.no/ dataverse/ntnu).

**Funding:** This work was supported by several grants. The presented work and also the validation of events from Cohort 1, 2 and 4 was supported by research grants from Nord-Trøndelag Hospital Trust (https://www.hnt.no/om-oss/contact/about-helse-nord-trondelag/) to C.S.R. and H.D., while the validation of events in Cohort 3 was supported by a research grant from SomaLogic (Inc., Boulder, CO, USA; https://somalogic.com/) to M.A.W. (Olson KA, Beatty AL, Heidecker B, Regan MC, Brody EN, Foreman T, et al. Association of growth differentiation factor 11/8, putative anti-ageing factor, with cardiovascular outcomes and overall mortality in humans: analysis of the Heart and Soul and HUNT3 cohorts. Eur Heart J. 2015;36 (48):3426-34. Epub 2015/08/22. doi: 10.1093/ eurheartj/ehv385. PubMed PMID: 26294790; PubMed Central PMCID: PMCPMC4685178.). The validation of events from Cohort 5 was funded by research grant to B.O.Å. the Research Council of Norway (grant number 231149/F20; https://www. forskningsradet.no/en/). None of the funders had any role in neither designing the study, collecting or analyzing the data, decision to publish, nor preparation of the manuscript.

**Competing interests:** The authors have declared that no competing interests exist.

infarction and heart failure where most valid when placed at cardiology units, while ischemic stroke when discharged from an internal medicine unit.

## Conclusions

The diagnoses of acute myocardial infarction and stroke had excellent validity when placed during hospital stays. Similarly, heart failure diagnoses had excellent validity when echocardiography was performed before placing the diagnosis, while overall the diagnoses of heart failure and transient ischemic attack were less valid. In conclusion, the results indicate that cardiovascular diagnoses based on objective findings such as acute myocardial infarction and stroke have excellent validity and may be used as endpoints in clinical epidemiological studies with less rigid validation.

## Introduction

Mortality is considered the most reliable endpoint in clinical studies [1], while cardiovascular clinical outcomes as acute myocardial infarction (AMI), heart failure (HF), and stroke are commonly used in cardiovascular epidemiological studies [2,3]. Such epidemiological studies have provided important knowledge on how risk factors influence diseases and well-being [2,4–6].

Hospital medical records provide an easy and affordable access for researchers to cardiovascular endpoints. By linking hospital endpoints with data from epidemiological studies, large-scale studies of causative relations between risk factors and diseases may be performed at a low cost. In such studies, the reliability of the chosen endpoints is crucial and the clinical impact depends on the validity of both the exposure and outcome variables. Thus, to ensure high-quality science, the outcome variables must be accurate and unbiased [1].

Multiple publications have addressed the validity of different cardiovascular diagnoses [7–13]. According to a systematic review including 81 studies, most studies included just a limited number of participants (n <400) [7]. The same review showed a considerable heterogeneity between the estimated positive predictive values (PPVs) for different cardiovascular diagnoses, and the variability was even more pronounced when evaluating sub-categories of the cardiovascular diagnoses. The validity of diagnoses according to the position (primary versus subsequent diagnoses) is scarcely evaluated [7]. Moreover, outpatient status may influence diagnostic validity, as the diagnostic work-up and observation time may differ from hospitalized patients. Still, only a few studies have evaluated if the diagnostic validity differ between in- and outpatients and the published results are conflicting [13–15]. Another characteristic, which may influence diagnostic validity, is the unit placing the diagnosis. Some have found a gradient in PPV of cardiac diagnoses across units, with highest PPVs for cardiology units lowest PPVs for non-medical units [8,16–18].

It is commonly recommended to validate hospital endpoints before scientific use [1,12,16,18], but there is limited data on the factors influencing the validity of cardiovascular diagnoses. The aim of this study was to evaluate how diagnoses-specific characteristics like type, sub-categories, and position of cardiovascular diagnoses specified as AMI, HF, and stroke or cerebral transient ischemic attack (TIA) affected the precision level across a Norwegian hospital trust. Secondary aims were to assess how patient- and hospital characteristics, as sex,

inpatient vs. outpatient status, and the unit placing the diagnoses affected the validity of the same diagnoses.

## Methods

Design: Validation study evaluating predictors of diagnostic precision across hospital diagnoses of AMI, HF, stroke, and TIA.

### Study populations

The study population consisted of five cohorts validated for recent and ongoing cohort studies [2,3,19] related to the Trøndelag Health Study (HUNT) [20]. HUNT is a large population-based cohort study performed across the Nord-Trøndelag region, Norway. The region hosts two local hospitals in Levanger and Namsos, respectively. These hospitals together form Nord-Trøndelag Hospital Trust (HNT). Cohort 1 and 2 was considered quality improvement projects, without the need for consent according to Norwegian law. Patients included in cohort 3–5 gave written consent during inclusion in the HUNT survey. The study was conducted according to the principles expressed in the Declaration of Helsinki, and approved by the Regional Ethics Committee for Medical and Health Research Ethics (REK 2021/230889).

**Cohort 1:** A random sample (n = 224) diagnosed with AMI in any HNT hospital during 2009–2016.

**Cohort 2**: All cases (n = 539) diagnosed with AMI at Levanger Hospital during 2009–2010.

**Cohort 3**: Selected cases (n = 1325) with a diagnosis of AMI, HF, stroke/TIA in any HNT hospital between January 2007 and January 2014 validated to match the population from a Heart and Soul sub-study [21].

**Cohort 4:** All participants (n = 3407) of the second and third wave of the HUNT Study [20] diagnosed with HF in any HNT hospital between October 2006 and February 2018.

**Cohort 5**: All women (n = 2888) included in the Norwegian Medical birth registry (after its launch in 1967) which also participated in the first, second and third wave of the HUNT Study [22] and were diagnosed with AMI, HF, stroke, or TIA in any HNT hospital between September 1987 and June 2015.

As each patient could be included in more than one cohort and diagnosis category, we merged all cohorts based on the subject specific national identification numbers. In cases with repetitive validations or events, we compared diagnosis types, positions, origin units and dates. Subsequently, duplets were excluded. Cohort 1 included only one AMI diagnosis per patient, while all available AMI diagnoses were evaluated in Cohort 2. In Cohort 3 just the first event of each main diagnosis category was included, and similarly just one HF diagnosis per patient was included in Cohort 4. In Cohort 5 up to three separate events of each diagnoses category were included.

### Data collection

We used the patient administrative system (PAS) to identify the diagnoses of interest specified in Table 1. Both primary and subsequent diagnoses were included. Cohort 1, 2 and 4 was scrutinized for not previous listed cardiovascular diagnoses during validation for the index diagnosis, and events added in the proper cohort if all relevant criteria was met. For Cohort 3–5 the initial PAS search also included adjacent diagnoses that were validated according to the presence of AMI, HF, and stroke/TIA. The adjacent International Classification of Diseases (ICD) diagnoses were: i) for AMI; ICD-9 codes 412, 414, and 427.5, ICD-10 codes I25.1-.2, I46, Z95.1,.5, ii) for HF; ICD-9 codes 415, 416, 425, ICD-10 codes I11, I13, and I42, and iii) for cerebrovascular diseases ICD-10 codes G46, I65-67, I69. If the adjacent diagnosis met the

**Table 1. Overview of the study cohorts.**

| Cohort | N | Women, N (%) | Validated diagnoses | | Definition | Time frame |
|---|---|---|---|---|---|---|
| | | | ICD-9 | ICD-10 | | |
| **Cohort 1** *(AMI)* | 224 | 85 (37.9%) | - | I21-I22 | *a* | *2009–2016* |
| **Cohort 2** *(AMI)* | 539 | 180 (33.4%) | - | I21-I22 | *a* | *2009–2010* |
| **Cohort 3** | 1325 | 513 (38.7%) | - | | | |
| *AMI* | | | | I21-I22 | *a* | *2007–2014* |
| *Heart failure* | | | | I50 | *b, c* | |
| *Stroke* | | | | I61-I64 | *d* | |
| *TIA* | | | | G45; .0–3, .8 & .9 | *e* | |
| **Cohort 4** (Heart failure) | 3407 | 1737 (50.9%) | - | I50 | *c* | *2006–2018* |
| **Cohort 5** | 2888 | 2888 (100%) | | | | |
| *AMI* | | | 410–411 | I20-I22 | *a* | *1987–2015* |
| *Heart failure* | | | 428 | I50 | *b, c* | |
| *Stroke* | | | 430–434 | I60-I64 | *d* | |
| *TIA* | | | 435 | G45; .0–3, .8 & .9 | *e* | |
| **Unique participants** | 7164 | 4746 (66.2%) | | | | |

Validation was performed according to *a*) The third universal definition of MI [24], *b*) ESC (European Society of Cardiology) 2012 heart failure guidelines [25], *c*) ESC 2016 heart failure guidelines [26], *d*) Stroke was categorized as 1) ischemic strokes (cerebral infarctions verified by magnetic resonance imaging or computer tomography or no visible infarction but symptoms and signs lasting ≥24 hours) and 2) hemorrhagic stroke, *e*) TIA was defined as typical symptoms and signs lasting <24 hours with no evidence of new brain damage on cerebral imaging.

Abbreviations: AMI, acute myocardial infarction; ICD, International Classification of Diseases; N, numbers; TIA, transient ischemic attack.

diagnostic criteria for either AMI, HF, stroke, or TIA a valid diagnosis was stated. To ensure complete dataset of AMI diagnoses we scrutinized the regional myocardial infarction registry [23] for events not registered in PAS in Cohort 2.

Due to the challenge of confirming myocardial infarction as the cause of a cardiac arrest in the hospital records, these diagnoses were excluded. Further, cerebral events following trauma, intracerebral malignancy, transient global amnesia, and other non-ischemic or non-hemorrhagic primary etiologies were excluded. ICD-10 code I64: "stroke not specified as hemorrhage or infarction" were included as stroke but excluded from the analyses of sub-categories of cerebrovascular diseases. Amaurosis fugax and focal transitory neurologic diagnoses were included as stroke or TIA based on the documentation available if the criteria were fulfilled.

Outpatient status in Cohort 4 was stated in cases never hospitalized for HF. Some events in Cohort 5 (before (1967–1987) and during the study period (1987–2001)) were diagnosed prior to digitization of the medical records. These cases were validated by evaluation of the printed medical records. Thirteen (13) outpatients diagnosed before 1992 registered without a discharge unit were excluded from the sub-analyses.

Diagnoses specific factors such as type of diagnosis, diagnostic sub-categories, and position of the diagnoses were extracted directly from the PAS. The personal identification numbers indicating age and sex of the patients and hospital related factors such as admission status (inpatients vs. outpatients) and origin (hospital and unit) were also included. Medical records accessed in different steps between June 2021 and June 2023.

## Validation procedures

Five experienced HNT cardiologists (HD, TG, OK, KS, and BK) validated the diagnoses by manually reviewing hospital medical records. Cohort 1 was reviewed by the first cardiologist,

Cohort 2 by the first four, and Cohort 3–5 by the latter, except for the printed medical records in Cohort 5 which were validated by the first cardiologist. According to respective diagnostic criteria, all available information in the medical record files, including free text, electrocardiography, laboratory values, echocardiography, and radiologic examinations were used to dichotomize the diagnoses into valid or not valid (Table 1). Additional information as notes or referrals from other hospitals or out-of-hospital cardiologists were also used. In cases where the available information was insufficient for confirmation of a valid diagnosis, the diagnoses were classified as "not valid" for all Cohorts. Additionally, in Cohort 4 the diagnoses were classified as "uncertain" if they were considered likely valid but lacked some information to meet the criteria of a valid HF event, usually indicating no description of an echocardiographic evaluation. Uncertain HF cases were excluded from the analyses unless otherwise stated.

**AMI** diagnoses from Cohort 1–3 were sub-grouped into type 1–5 AMIs, in line with European Society of Cardiology (ESC) guidelines [24]. Type 1 indicates spontaneous myocardial infarction related to a coronary artery lesion causing intraluminal thrombus, decreased myocardial blood flow and necrosis. Type 2 represents imbalance between myocardial oxygen supply and demand, type 3 cardiac death due to myocardial infarction not fulfilling the type 1 criteria, whereas type 4 and 5 are AMIs associated with coronary intervention and surgery, respectively. Just 11 diagnoses were type 3–5 AMIs, and these were excluded from the sub-analyses. AMI diagnoses were further classified as ST-elevation myocardial infarction (STEMI) or non-ST-elevation myocardial infarction (NSTEMI) according to guidelines [24], except for events diagnosed by ICD-9 code 410, acute myocardial infarction, in Cohort 5 as this ICD-9 code does not differentiate between STEMI and NSTEMI. False negative AMIs were not further classified.

**HF** diagnoses included both acute and chronic HF. For Cohort 4, the most likely etiology of HF was specified (ischemic and non-ischemic where the latter included arrhythmia, metabolic, infiltrative, valvular, genetic/hypertrophic/dilated, and others). Several etiologies could be present in the same patient. Where possible, HF was classified into sub-categories of preserved ejection fraction (EF) (HFpEF), mildly reduced EF (HFmrEF), and reduced EF (HFrEF) according to guidelines [26].

**Cerebrovascular diseases** included stroke (ischemic, non-traumatic hemorrhagic, and unspecified) and TIA. Ischemic stroke was validated based on typical focal symptoms and signs combined with findings from cerebral imaging. Some patients received diagnoses for both hemorrhagic and ischemic stroke on the same date. In these cases, the diagnosis in the first position (primary diagnosis) was included as the first event, while subsequent diagnoses were included as additional events in the sub-analyses. TIA was validated based on typical symptoms and signs, even when not observed by a physician, lasting <24 hours and without evidence of new brain damage on cerebral imaging.

## Classification of hospital related factors

Inpatients were admitted to hospital and stayed for more than six hours. Outpatient visits included consultations at the hospitals lasting less than six hours. Cardiology outpatients underwent a consultation at the hospitals' cardiac outpatient clinics (general cardiology or HF clinics) or stayed at a cardiology unit for less than six hours.

The discharge unit was defined based on the allocation of patients. In patients treated at the intensive care units, we allocated the discharge unit corresponding to the physician in charge. Consultations in the emergency room were allocated to the department of internal medicine. Since the medical intensive care units were chaired by cardiologists, events discharged from these units were allocated to a cardiology unit. Levanger Hospital had a separate medical

stroke-unit, while no stroke unit was present at the other hospital. Thus, patients discharged from the stroke unit were allocated to the internal medicine unit.

## Statistics

Normally distributed continuous data are presented as mean and standard deviation (SD). Proportions are presented as *n* (%). We calculated the positive predictive value as the percentage of valid diagnoses divided by the sum of valid and non-valid diagnoses. Sensitivity was calculated as valid diagnoses divided by all patients with the specified disease. Differences between groups were compared using Chi Square test, two proportion z tests, or logistic regression, respectively. A p-value <0.05 was considered statistically significant. Analyses were performed using SPSS, versions 28 and 29 (IBM).

## Results

### Study population characteristics

A total of 8639 diagnoses from 7164 unique patients were included in the analyses (Table 1). Some patients were included in more than one Cohort. Table 2 shows the details according to first ever cardiovascular event, and thus, each patient is counted only once, leaving 5953 first events for unique patients. Cohort 5, consisting of only women, contributed approximately 40% of the total patients, and established a high proportion of women overall, particularly for cerebrovascular diseases. The HF group had the largest number of patients (46%), and included older patients, leading to an older overall population. For all Cohorts >90% were inpatients and most were discharged from the cardiology units.

### Validity and the characteristics of cardiovascular diagnoses

In Table 3 the excellent validity of the AMI diagnoses is shown by the high PPV and sensitivity (≥0.96 and >0.90, respectively). HF had lower PPV at 0.75 (absolute difference 0.21 compared to AMI, p <0.001), when only valid diagnoses were included as true positives. When the 439 events classified as "uncertain HF" were included as true positive events, PPV was 0.78. If the "uncertain HF" events were included as false positive, the PPV was only 0.67. Main diagnoses of HF had excellent sensitivity (>0.90). Cerebrovascular diagnoses had a high PPV of 0.83 while the overall sensitivity of 0.89 was somewhat lower than for HF (p <0.001) but not lower

**Table 2. Overview of patient- and hospital characteristics according to first events per patient of the main cardiovascular diagnosis.**

|  | Acute Myocardial Infarction (n = 1614) | Heart Failure (n = 3321) | Cerebrovascular Diseases (n = 1750) | Total (n = 5953) |
|---|---|---|---|---|
| Age, mean (SD), years | 67.4 (12.7) | 78.8 (11.5) | 62.2 (11.6) | 72.3 (14.0) |
| Female, n (%) | 876 (54.3%) | 1783 (53.7%) | 1498 (85.6%) | 3624 (60.9%) |
| Inpatients, n (%) | 1585 (98.2%) | 3010 (90.6%) | 1455 (83.1%) | 5377 (90.3%) |
| **Discharging unit** |  |  |  |  |
| Cardiology, n (%) | 1195 (74.0%) | 1675 (50.4%) | 129 (7.4%) | 2620 (44.0%) |
| Internal medicine, n (%) | 348 (21.6%) | 1203 (36.2%) | 1235 (70.6%) | 2518 (42.3%) |
| Others [a], n (%) | 70 (4.3%) | 440 (13.2%) | 379 (21.7%) | 802 (13.5%) |

Some patients were included in more than one Cohort. The table show data according to first ever cardiovascular event, and thus, each patient is counted only once, leaving 5953 first events for unique patients in the table.

[a] Surgical, orthopedic, obstetric, and gynecological unit.

Abbreviations: SD, standard deviation.

Table 3. The validity of different types of cardiovascular diagnoses overall and according to subgroups.

| | N events | True Positive Events | PPV | Sensitivity |
|---|---|---|---|---|
| **Acute myocardial infarction** | **2189** | **2097** | **0.958** | **0.905** |
| STEMI | 796 (36.4%) | 777 | 0.976 | 0.999 |
| NSTEMI | 946 (43.2%) | 907 | 0.959 | 0.999 |
| AMI Type 1 | 859 (39.2%) | 851 | 0.991 | - |
| AMI Type 2 | 104 (4.8%) | 103 | 0.990 | - |
| **Heart failure**[a] | 3587[a] | 2694 | 0.751 | 0.930 |
| Ischemic | 1153 (32.1%) | 1148 | 0.996 | 0.980 |
| Non-ischemic | 1712 (47.7%) | 1712 | 1.000 | 0.939 |
| **Cerebrovascular diseases** | 2012 | 1678 | 0.834 | 0.894 |
| Ischemic Stroke | 1034 | 898 | 0.868 | 0.793 |
| Hemorrhagic Stroke | 257 | 205 | 0.798 | 0.748 |
| TIA | 926 | 516 | 0.557 | 0.860 |

[a] Valid events only, excluding 439 «uncertain» events. Only valid HF diagnoses were classified into subgroups of etiology.

Abbreviations: STEMI, ST-elevation myocardial infarction; NSTEMI, Non-ST-elevation myocardial infarction; PPV, positive predictive value, otherwise as in Table 1.

than for AMI (p = 0.255). A significant part of stroke diagnoses (n = 103) was misclassified as TIA (false negative events).

## Validity of sub-categories of cardiovascular diagnoses

Sub-categories of AMI had somewhat higher PPV than the main diagnoses. The differences were statistically significant only for STEMI and type 1 AMIs (p ≤0.02) (Table 3). Similarly, HF diagnoses classified according to etiology had better validity and less false positives than HF diagnosis lacking etiological details in the medical record (absolute difference 0.25, p <0.001). Sensitivity for both AMI and HF remained excellent (>0.90) in all sub-categories. Echocardiography was performed in 2234 (62.3%) of the patients with any HF diagnosis. In this group, the PPV was 0.97 and sensitivity was 0.94 (Fig 1), with only 63 false positive and 12 uncertain HF diagnoses. Out of the 2213 patients stratified according to EF, we found 36 false positive in the HFrEF group, 22 in the HFmrEF group and 2 in the HFpEF group.

For cerebrovascular sub-categories, ischemic stroke had higher PPV (0.87) than hemorrhagic stroke, and TIA (PPVs 0.80 and 0.56, respectively). The sub-categories of cerebrovascular diseases had sensitivity ranging 0.75–0.86, all being lower than the main diagnoses (p ≤0.021). The ICD-10 code I64, "Stroke, not specified as hemorrhage or Infarction" was used only 25 times whereof all were valid (11 were classified as hemorrhagic and 14 as ischemic during validation).

**The importance of the position of the diagnoses.** Fig 2 shows the PPVs according to the position of the diagnoses. Primary diagnoses had significantly higher PPVs for presented cardiovascular diagnoses. Except for AMI, the absolute differences in PPV between primary and subsequent diagnoses were ≥0.15 (SD 0.06). Subsequent position of the HF diagnoses was common with 2254 events compared to 1333 events with HF as the primary diagnosis. For AMI and cerebrovascular diseases subsequent positioning were rare and occurred in 6–21%.

**Validity of diagnoses related to patient- and hospital characteristics.** Females had lower validity of AMI, HF and TIA diagnoses compared to males with absolute differences in PPV of 0.04, 0.06 and 0.23, respectively, all p <0.001). For the sub-categories of stroke, the differences were not significant (p ≥0.08). The PPVs were higher in women for hemorrhagic stroke (absolute PPV difference 0.11) and lower for ischemic stroke (absolute PPV difference 0.05).

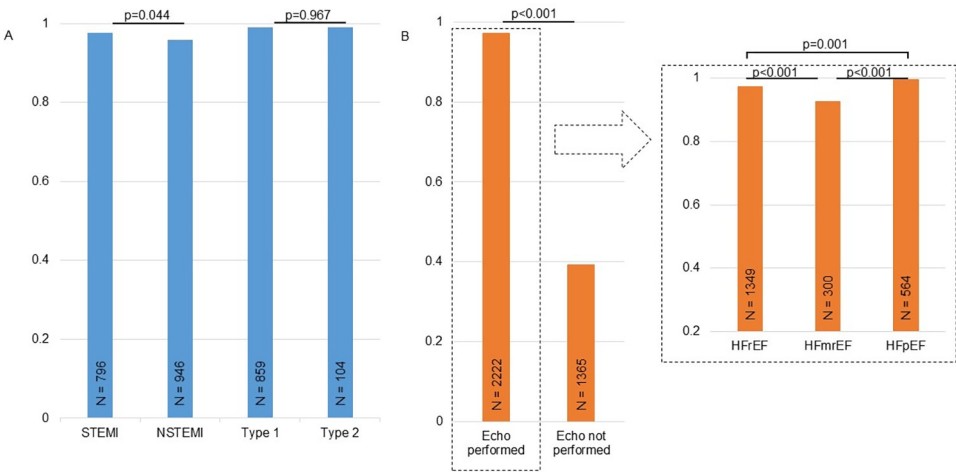

**Fig 1. The diagnostic validity of sub-categories of AMI and HF.** The positive predictive values of different sub-categories of AMI (blue) and HF (orange). The counts are provided inside the columns. P-values for comparison of groups are highlighted on top of the columns.

Fig 3 shows that the PPVs for all evaluated cardiovascular diagnoses were higher in inpatients compared to outpatients. Additionally, outpatient status was infrequent among the validated diagnoses being present in 36 (1.7%) of AMI events, 389 (10.8%) of HF events, 101 (7.8%) of stroke events, and 286 (30.8%) of TIA events, respectively.

Fig 4 shows the importance of the units placing the diagnoses for the PPVs. For AMI and HF the PPVs were highest when the diagnoses were placed by a cardiology unit (PPVs 0.97 and 0.90, respectively) and lowest PPVs when placed by non-medical units (0.88 and 0.46), both p <0.001. For stroke PPV was >0.83 for cardiology and internal medicine units, while fairly lower for non-medical units with PPV ≤0.77. For TIA the corresponding PPVs ranged 0.53 to 0.61 for the different units, with no statistically significant differences (p ≥0.157).

Comparing data from the two hospitals, we found no significant differences in PPVs for AMI, stroke, or TIA, p ≥0.081. HF diagnoses had a higher validity at Levanger Hospital with PPV 0.83 compared to Namsos Hospital with PPV 0.61 (p <0.001), and similarly,

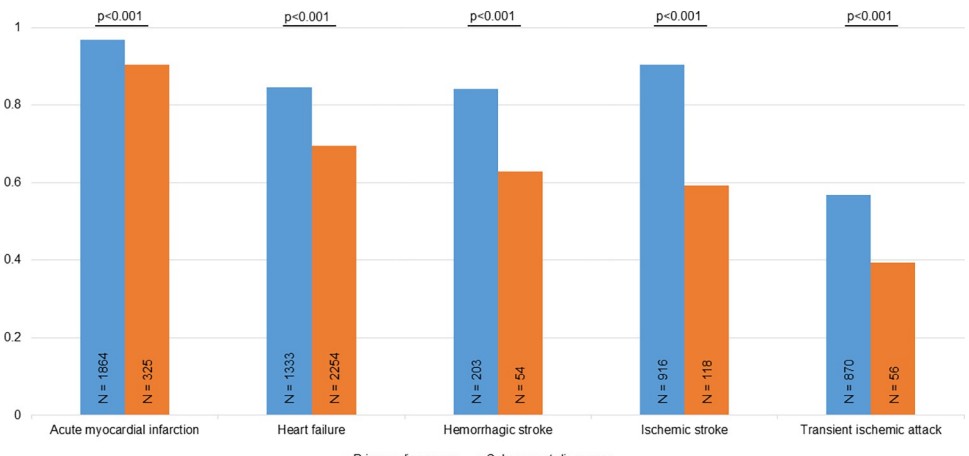

**Fig 2. The diagnostic validity of primary and subsequent cardiovascular diagnoses.** The positive predictive values of the cardiovascular diagnoses are shown specified for primary and subsequent diagnoses. Explanations as in Fig 1.

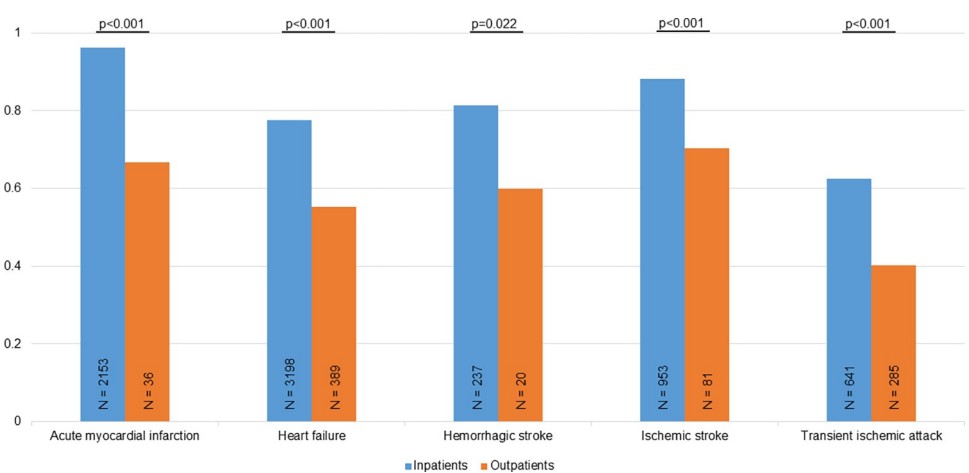

**Fig 3. The diagnostic validity of cardiovascular diagnoses according to in- and outpatient status.** Explanations as in Fig 1.

echocardiography was more often performed before a HF diagnosis was placed at Levanger Hospital compared to Namsos Hospital (60% vs. 42%, p <0.001), respectively.

## Discussion

In this large-scale validation study, we found excellent validity for diagnoses of AMI and ischemic stroke when placed as primary diagnoses with PPVs ≥0.90. Diagnoses of HF and TIA had a modest validity (PPVs 0.75 and 0.56), except for when echocardiography was performed prior to placing a HF diagnosis. Subgroups of diagnoses were more valid than main diagnoses except for hemorrhagic stroke and TIA. Diagnoses placed during hospital stays were consistently more valid than diagnoses placed in the outpatient settings, with an absolute difference in PPVs of ≥0.18 (mean difference 0.23) for all validated diagnoses. Moreover, the diagnostic validity of cardiac diagnoses was higher when placed by cardiology units compared to non-cardiology units.

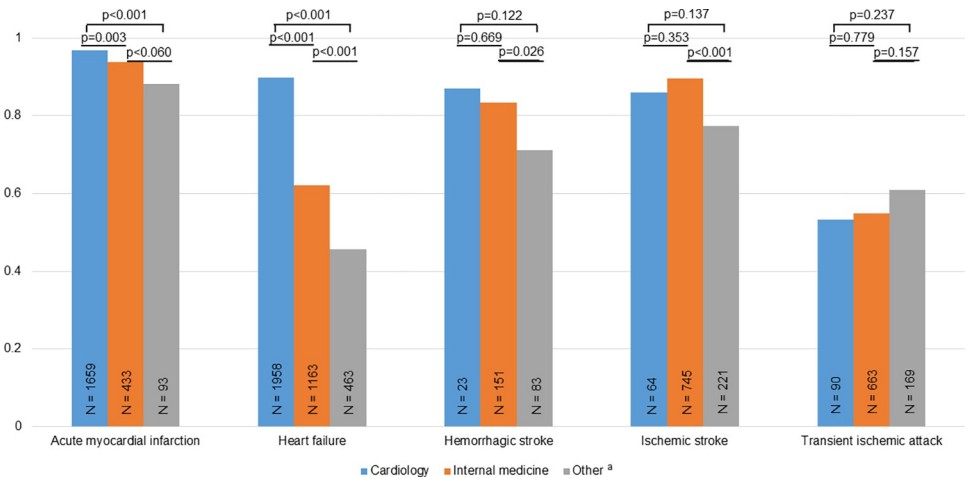

**Fig 4. Positive predictive value of cardiovascular diagnoses by discharging units.** Explanations: [a] Surgical, orthopedic, obstetric, and gynecological units, otherwise as in Fig 1.

The presented findings of the high to excellent validity of diagnoses of AMI, stroke, and HF when placed in inpatients or where an echocardiography was performed before placing a HF diagnosis adds significantly to the literature. This finding may strengthen the scientific society's interpretation of findings from large population studies even when validation of the diagnoses is not performed. Further, the modest validity of diagnoses of TIA and HF (when not further supported by echocardiography) indicate that these diagnoses may introduce more scientific uncertainty when used in epidemiological studies without proper validation.

## The validity of cardiovascular diagnoses

The finding of a high validity for diagnosing AMI and stroke are in line with previous publications [7,27–30]. Similarly, the findings of moderate validity for HF are also known [7,8,13,16]. A higher validity for stroke than TIA is previously shown by others [29,31]. The non-discriminating symptoms of HF [25,32] and lack of objective findings in TIA may contribute to a modest validity for these diagnoses [31].

In those where the diagnoses of AMI and HF were classified into sub-categories, both PPV and sensitivity were higher. Previous research on sub-categories of both AMI and HF are scarce, while it is previously shown that echocardiography augments the validity of HF diagnoses [8,16,17]. Sub-categories of stroke showed unaltered PPVs and a modest reduction in sensitivity compared to the main diagnosis, supporting previous findings [29,30]. In a recent paper, Øie et al. evaluated the validity of newer cerebrovascular disease diagnoses and found higher PPV for all intracranial hemorrhage and its subgroups in inpatients, while among outpatients the PPVs were in line with our result [14].

In line with previous studies, the presented study indicates better validity when the cardiovascular diagnoses were positioned as the primary diagnoses, compared to subsequent positioning [7,13,16,17,30]. Even though subsequent positioning of the cardiovascular diagnoses, as well as outpatient status, were present only in a minority of the validated events both were closely associated with reduced validity. Further, except for TIA, the validity was higher when the diagnosis originated from a unit with the most relevant specialists. For TIA, no sub-classification based on patient- or hospital characteristics had PPVs ≥0.63, making it the least valid diagnosis in our study.

The finding of excellent validity of HF diagnoses when combined with up-front echocardiography is supported by Schaufelberger et al. [8] who found that 97% of cases with "definite" HF had an echocardiography performed. This highlight the need for comprehensive evaluation to reveal objective findings of cardiac pathology before a valid diagnosis of HF can be placed. According to the ESC guidelines, echocardiography is a cornerstone in HF diagnostics [25,26], and the presented results indicate that echocardiography is important for a valid HF diagnoses. Importantly, the presented findings indicate that if HF diagnoses are not comprehensively validated, or echocardiography was not performed up-front, the HF diagnoses have too low validity to be used as endpoints in population studies. Further, when the HF diagnoses were placed in one of the two cardiology units the PPVs were ≥0.89, compared to the modest PPVs of ≤0.70 when the diagnoses were placed in non-cardiology units or for subsequent diagnoses.

The findings of more valid diagnoses in inpatients vs. outpatients are not surprising. PPV is by definition positively associated with the prevalence of the disease in the population studied [33]. The same mathematical relation may also have influenced the higher validity when the cardiac diagnoses were placed in cardiology units compared to non-cardiology units. Moreover, the PPVs are expected to be higher in inpatients, as the patients are more likely to receive more examinations and longer observation time during the hospital stay. In addition, some

outpatients may have received misplaced diagnoses of HF or stroke at follow-up, instead of more vague observational diagnoses, which may have impaired the validity for outpatients. However, our results show that diagnoses including objective findings (e.g. rise and fall of troponins, electrocardiography changes, echocardiographic findings, and ischemic or hemorrhagic areas on cerebral imaging) have higher validity compared to those based on medical history and clinical findings. The low precision and agreement of medical history and clinical findings are well known from the literature but may be forgotten in clinical practice [25,26,31].

## Strengths and limitations

The main strengths of our study are the large sample and long time-span of comprehensively validated diagnoses. A total of 8639 cardiovascular events were included, with more than 2000 AMIs, 4000 HF diagnoses, and more than 2000 cerebrovascular events. Thus, our study includes far more events than comparable studies [7,12,13,27]. The same electronic medical record was used across the two hospitals, and electrocardiograms, laboratory tests, imaging procedures, and reports were available during validation. Further, emigration has for decades been low in the current area and no private hospitals are located in the hospital trust. A wide range of physicians at different levels of experience placed the diagnoses over decades. Even though cohorts 3–5 were identified based on participation in the HUNT Study, we expect that the high participation rate in HUNT limits the potential selection bias in these cohorts. Thus, we believe the presented results could be broadly generalized to other hospitals where cardiac diagnoses are placed in cardiac, medical, and non-medical units.

The large sample size enabled comprehensive sub-analyses evaluating the importance of diagnostic sub-groups, patient characteristics and diagnostic units. Still, some sub-groups included only a few participants. Also, some sub-categories were not included in the analyses due to a very low number of participants or data not available in the patient archive system. In particular, this challenged the evaluation of subsequent diagnoses and outpatients.

Cohort 5, a women only cohort, made the participation rate of women higher than commonly presented [34]. As symptoms of cardiovascular diagnoses, especially in elderly women, are different from men this may have biased the results towards a lower validity [35]. However, women are underrepresented in cardiovascular clinical trials, making this study even more important [36].

A significant limitation in our study is that validation was performed only once and only by cardiologists. This may have led to a systemic bias. As Cohort 1, 2 and 4 were scrutinized for not previous listed cardiovascular diagnoses and adjacent diagnoses, some false negative diagnoses were found, but the study design did not allow for full exploration of false negative events. This constitute a risk of overestimating the sensitivity of the diagnostic procedures. Some sub-groups included few events presenting a risk of type 2 errors [37]. Still, for most analyses the large number of patients counteracts this problem and provide adequate statistical power. Re-validation by other clinical experts would have strengthened the results.

Retrospective review of existing medical records has some limitations, like missing data or the lack of exhaustive examinations- or description of such. Excessive or busy workload may also contribute to errors in data entry or limited documentation. Both factors constitute challenges to validation. This was seen by the category "uncertain" HF diagnoses, where the most common limitation during the validation was missing important information, usually echocardiographic evaluation. An important message to improve the validity of clinical diagnoses in the future includes careful attention when placing diagnoses not supported by objective findings, as well as research to better identify diseases where objective findings are not mandatory as illustrated by the TIA diagnoses in this work.

## Conclusions

The diagnoses of AMI and stroke had excellent validity when placed during hospital stays. Similarly, HF diagnoses had excellent validity when echocardiography was performed before placing the diagnosis, while overall the diagnoses of HF and TIA were less valid. The diagnoses accompanied with details of underlying diagnostic sub-groups had even higher PPVs and diagnoses in primary position had higher validity than subsequent positions. The cardiovascular diagnoses were more valid in inpatients and when placed by relevant specialists. In conclusion, the results indicate that cardiovascular diagnoses based on objective findings such as AMI and stroke have excellent validity and may be used as endpoints in clinical epidemiological studies with less rigid validation. For HF and TIA the lower validity will introduce some uncertainty when used as study endpoints, hence validation or additional information supporting the diagnoses is warranted.

## Acknowledgments

We acknowledge the important contributions of the cardiologists Torbjørn Graven, Olaf Kleinau, Kyrre Skjetne and Bjørnar Klykken during the validation procedures.

## Author Contributions

**Conceptualization:** Håvard Dalen.

**Investigation:** Cathrine Sæthern Rye.

**Project administration:** Håvard Dalen.

**Supervision:** Håvard Dalen.

**Validation:** Håvard Dalen.

**Writing – original draft:** Cathrine Sæthern Rye.

**Writing – review & editing:** Anne Pernille Ofstad, Bjørn Olav Åsvold, Pål Richard Romundstad, Julie Horn, Håvard Dalen.

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
