## [Decision Letter · Decision Letter 0]

9 Jan 2024

PONE-D-23-36180The influence of diagnostic subgroups, patient- and hospital characteristics for the validity of cardiovascular diagnoses – Data from a Norwegian hospital trusPLOS ONE

Dear Dr. Rye,

Thank you for submitting your manuscript to PLOS ONE. After careful consideration, we feel that it has merit but does not fully meet PLOS ONE’s publication criteria as it currently stands. Therefore, we invite you to submit a revised version of the manuscript that addresses the points raised during the review process.

We look forward to receiving your revised manuscript.

Kind regards,

Vikramaditya Samala Venkata

Academic Editor

PLOS ONE

Journal Requirements:

2. In the online submission form, you indicated that "Data cannot be shared publicly to protect participants privacy, due to containing

personal identifiable and sensitive data. (European GDPR rules).

Data are available from the Helse Nord Trøndelag Institutional Data Access, and may be shared upon reasonable request for researchers who meet the criteria for access to confidential data."

Additional Editor Comments:

Article is well written, just needs a few questions answered as noted by reviewers below. Please refer to the reviewer comments below

Reviewers' comments:

Reviewer's Responses to Questions

**Comments to the Author**

1. Is the manuscript technically sound, and do the data support the conclusions?

Reviewer #1: Yes

Reviewer #2: Yes

2. Has the statistical analysis been performed appropriately and rigorously? 

Reviewer #1: Yes

Reviewer #2: Yes

3. Have the authors made all data underlying the findings in their manuscript fully available?

Reviewer #1: Yes

Reviewer #2: Yes

4. Is the manuscript presented in an intelligible fashion and written in standard English?

Reviewer #1: Yes

Reviewer #2: Yes

5. Review Comments to the Author

Reviewer #1: The manuscript "The influence of diagnostic subgroups patient- and hospital characteristics for the validity of cardiovascular diagnoses – Data from a Norwegian hospital trust" presents a comprehensive analysis of the validity of cardiovascular diagnoses, specifically acute myocardial infarction (AMI), heart failure (HF), and cerebrovascular diseases, in a hospital setting. The study involving 7164 patients aims to understand how different factors affect the accuracy of these diagnoses, which is crucial for epidemiological studies and patient care.

Technical Soundness and Data Support:

Statistical Analysis: The use of positive predictive values (PPV) and sensitivity as key metrics is appropriate for this study. The presentation of results, including the specific PPVs for different conditions and patient situations (inpatient vs. outpatient, primary vs. subsequent diagnoses), provides a detailed understanding of the factors influencing diagnostic accuracy. It seems that the statistical methods used, including Chi Square tests and logistic regression, are suitable for the data and research questions.

Data Interpretation: The conclusion that diagnoses of AMI and stroke have high validity, especially when made during hospital stays, and that HF diagnoses are most valid when supported by echocardiography, is well-supported by data. These findings are significant and may have implications for the use of hospital discharge diagnoses in epidemiological studies.

Data Availability and Ethical Compliance:

Data Restrictions: It is noted that there are some restrictions on the availability of data. While it is understandable that patient privacy and compliance with GDPR rules might limit data sharing, this could potentially restrict the ability for independent verification of results. It would be beneficial if the data could be made available in a de-identified format to other researchers upon reasonable request.

Ethical Compliance: The study's approval by the Regional Committee for Medical and Health Research Ethics of Mid-Norway and adherence to the Declaration of Helsinki are commendable. This ensures that the study meets high ethical standards.

Manuscript Presentation:

Clarity and Language: The manuscript is well-written and presents the research in a clear and understandable manner. The use of standard English and clear articulation of the research methods and findings makes the manuscript accessible to readers.

Competing Interests and Funding: The declaration that there are no competing interests is noted. It's important for maintaining the objectivity of the study. However, information regarding specific funding sources or grants, if any, was not mentioned. Including this information would provide transparency about the potential influences on the research.

Recommendations:

Data Sharing: Consider providing a more detailed explanation of the data restrictions and potential ways researchers can access the data while still complying with GDPR and privacy concerns.

Sub-Categories Analysis: Further elaboration on the validity of different sub-categories of diagnoses would be beneficial. This could provide deeper insights into which specific types of diagnoses are more or less reliable.

Future Research Directions: Suggestions for future research, especially on how to improve the validity of diagnoses with lower PPV like TIA, would be beneficial. This could guide subsequent studies in this area.

Reviewer #2: Excellent article that is very well written. Results are well presented and conclusions including limitations are well discussed. A couple of minor points

- One of the limitations is that this study is applicable mainly to the HUNT system and demonstrates the fidelity of their diagnoses. Unsure of the generalizability across other study groups

- As mentioned in the limitation section, ideally all cases should have been reviewed by at least two cardiologists if not more. This does limit the validity of conclusions slightly. Consider a repeat study in the future with better validation if possible

6. PLOS authors have the option to publish the peer review history of their article (what does this mean?). If published, this will include your full peer review and any attached files.

Reviewer #1: **Yes: **HARPREET SINGH GREWAL

Reviewer #2: **Yes: **Venkata Buddhavarapu

---

## [Author Response · Author response to Decision Letter 0]

22 Feb 2024

PONE-D-23-36180: The influence of diagnostic subgroups, patient- and hospital characteristics for the validity of cardiovascular diagnoses – Data from a Norwegian hospital trust.

We thank the Editor and the reviewers for their thoughtful comments and efforts to improve the manuscript. The constructive feedback from both reviewers is much appreciated. We have considered the reviewer comments carefully and made changes accordingly. Please see our replies below the comments. We hope that the revised manuscript will have sufficient priority for publication in PLOS ONE. 

Authors’ response to the point raised by the Academic Editor 

1. Style and formatting requirements: To the best of our knowledge, the main body of the manuscript and the title page are edited and formatted according to PLOS ONE’s style template. 

2. Data availability: Based on your kind feedback and PLOS journals requirements, we have now ensure that all underlying data used in the manuscript will be made freely available from the repository of Norwegian University of Science and Technology (DataverseNO) under the CC0 (Creative Commons Zero) license.

3. Reference list: The references has now been revised accordingly. 

Review Comments by Reviewer #1

The manuscript "The influence of diagnostic subgroups patient- and hospital characteristics for the validity of cardiovascular diagnoses – Data from a Norwegian hospital trust" presents a comprehensive analysis of the validity of cardiovascular diagnoses, specifically acute myocardial infarction (AMI), heart failure (HF), and cerebrovascular diseases, in a hospital setting. The study involving 7164 patients aims to understand how different factors affect the accuracy of these diagnoses, which is crucial for epidemiological studies and patient care.

Technical Soundness and Data Support:

Statistical Analysis: The use of positive predictive values (PPV) and sensitivity as key metrics is appropriate for this study. The presentation of results, including the specific PPVs for different conditions and patient situations (inpatient vs. outpatient, primary vs. subsequent diagnoses), provides a detailed understanding of the factors influencing diagnostic accuracy. It seems that the statistical methods used, including Chi Square tests and logistic regression, are suitable for the data and research questions.

Data Interpretation: The conclusion that diagnoses of AMI and stroke have high validity, especially when made during hospital stays, and that HF diagnoses are most valid when supported by echocardiography, is well-supported by data. These findings are significant and may have implications for the use of hospital discharge diagnoses in epidemiological studies.

Authors’ response:

Thank you for the comprehensive evaluation. We have no further comments to the above listed comments by reviewer 1.

Data Availability and Ethical Compliance:

Data Restrictions: It is noted that there are some restrictions on the availability of data. While it is understandable that patient privacy and compliance with GDPR rules might limit data sharing, this could potentially restrict the ability for independent verification of results. It would be beneficial if the data could be made available in a de-identified format to other researchers upon reasonable request.

Authors’ response:

We have now ensured that all underlying data used in the manuscript will be made freely available from the research repository of Norwegian University of Science and Technology (DataverseNO) under the CC0 (Creative Commons Zero) license.

Ethical Compliance: The study's approval by the Regional Committee for Medical and Health Research Ethics of Mid-Norway and adherence to the Declaration of Helsinki are commendable. This ensures that the study meets high ethical standards.

Authors’ response:

No further comments.

Manuscript Presentation:

Clarity and Language: The manuscript is well-written and presents the research in a clear and understandable manner. The use of standard English and clear articulation of the research methods and findings makes the manuscript accessible to readers.

Authors’ response:

No further comments.

Competing Interests and Funding: The declaration that there are no competing interests is noted. It's important for maintaining the objectivity of the study. However, information regarding specific funding sources or grants, if any, was not mentioned. Including this information would provide transparency about the potential influences on the research.

Authors’ response:

We have now included complete information of funding sources in both the PLOS ONE submission system and cover letter. 

The following text is included under “Funding Statement” in the submission system: 

This work was supported by several grants. The presented work and also the validation of events from Cohort 1, 2 and 4 was supported by research grants from Nord-Trøndelag Hospital Trust (https://www.hnt.no/om-oss/contact/about-helse-nord-trondelag/) to C.S.R. and H.D., while the validation of events in Cohort 3 was supported by a research grant from SomaLogic (Inc., Boulder, CO, USA; https://somalogic.com/) to M.A.W. (Olson KA, Beatty AL, Heidecker B, Regan MC, Brody EN, Foreman T, et al. Association of growth differentiation factor 11/8, putative anti-ageing factor, with cardiovascular outcomes and overall mortality in humans: analysis of the Heart and Soul and HUNT3 cohorts. Eur Heart J. 2015;36(48):3426-34. Epub 2015/08/22. doi: 10.1093/eurheartj/ehv385. PubMed PMID: 26294790; PubMed Central PMCID: PMCPMC4685178.). The validation of events from Cohort 5 was funded by research grant to B.O.Å. the Research Council of Norway (grant number 231149/F20; https://www.forskningsradet.no/en/). None of the funders had any role in neither designing the study, collecting or analyzing the data, decision to publish, nor preparation of the manuscript.

Recommendations:

Data Sharing: Consider providing a more detailed explanation of the data restrictions and potential ways researchers can access the data while still complying with GDPR and privacy concerns.

Authors’ response:

Please see our reply above. Data will be made freely available from the research repository (DataverseNO) under the CC0 (Creative Commons Zero) license.

Sub-Categories Analysis: Further elaboration on the validity of different sub-categories of diagnoses would be beneficial. This could provide deeper insights into which specific types of diagnoses are more or less reliable.

Authors’ response:

In this manuscript we chose to focus on factors easily extracted from the patient administrative system, and thus, easily available for researchers using cardiovascular diagnoses as clinical endpoints. 

As shown in the manuscript page 9-10 only a few patients were diagnosed with subcategories available from the patient archive system not included in the analyses, e.g. 11 patients with AMI type 3-5. One important finding was that the specific diagnoses, i.e. including information on subcategories, tended to be more valid than the more general diagnoses. We have now included some information in the limitation section highlighting that further evaluation of subcategories was not performed related to a low number of available diagnoses or information lacking in the patient archive system (page 20, lines 428-9).

Future Research Directions: Suggestions for future research, especially on how to improve the validity of diagnoses with lower PPV like TIA, would be beneficial. This could guide subsequent studies in this area.

Authors’ response:

We agree that further research on how to improve the validity of cardiovascular diagnoses, especially those with low positive predictive values like TIA, would be important. We have included some comments on this topic in the revised manuscript (page 20-21, lines 447-50).

Review Comments by Reviewer #2

Excellent article that is very well written. Results are well presented and conclusions including limitations are well discussed. A couple of minor points

- One of the limitations is that this study is applicable mainly to the HUNT system and demonstrates the fidelity of their diagnoses. Unsure of the generalizability across other study groups

Authors’ response:

As shown in the manuscript (page 5, lines 110-2) population 1 and 2 were included outside the HUNT Study, while the larger cohorts 3-5 were identified based on participation in the HUNT Study. However, the participation rate in the different waves of the HUNT Study has been high compared to other population studies. In details: 

HUNT1 had 77,212 participants (participation rate 89.4%)

HUNT2 had 65,237 participants (participation rate 69.5%)

HUNT3 had 50,807 participants (participation rate 54.1%)

HUNT4 had 56,078 participants (participation rate 54%)

Even though inhabitants not participating in the HUNT Study has been shown to have lower socioeconomic status, higher mortality and higher prevalences of several chronic diseases (Langhammer, et al. BMC Med Res Methodol 2012. Doi: 10.1186/1471-2288-12-143), we have no indication that the participation in the populational study influence the diagnostic procedures in the hospitals. Nevertheless, we believe that the large number of included patients in HUNT studies and the high participation rate reduce the selection bias, and that the presented data are generalisable to others. We have included some comments in the limitation section (page 19, lines 419-21).

- As mentioned in the limitation section, ideally all cases should have been reviewed by at least two cardiologists if not more. This does limit the validity of conclusions slightly. Consider a repeat study in the future with better validation if possible

Authors’ response:

We completely agree that for the best validity of the results all cases should ideally have been reviewed by at least two different clinical experts. However, in line with other similar studies such a strict methodology is not comon as the workload and available resources rarely are matched. The use of different experts for the validation procedures reduces its limitation, but the limitation still exists. We have included a comment in the limitation section of the revised manuscript (page 20, lines 441-2).

---

## [Decision Letter · Decision Letter 1]

28 Mar 2024

The influence of diagnostic subgroups, patient- and hospital characteristics for the validity of cardiovascular diagnoses – Data from a Norwegian hospital trus

PONE-D-23-36180R1

Dear Dr. Rye,

We’re pleased to inform you that your manuscript has been judged scientifically suitable for publication and will be formally accepted for publication once it meets all outstanding technical requirements.

Kind regards,

Vikramaditya Samala Venkata

Academic Editor

PLOS ONE

Additional Editor Comments (optional):

Reviewers' comments:

Reviewer's Responses to Questions

**Comments to the Author**

1. If the authors have adequately addressed your comments raised in a previous round of review and you feel that this manuscript is now acceptable for publication, you may indicate that here to bypass the “Comments to the Author” section, enter your conflict of interest statement in the “Confidential to Editor” section, and submit your "Accept" recommendation.

Reviewer #1: All comments have been addressed

Reviewer #2: All comments have been addressed

2. Is the manuscript technically sound, and do the data support the conclusions?

Reviewer #1: Yes

Reviewer #2: Yes

3. Has the statistical analysis been performed appropriately and rigorously? 

Reviewer #1: Yes

Reviewer #2: Yes

4. Have the authors made all data underlying the findings in their manuscript fully available?

Reviewer #1: Yes

Reviewer #2: Yes

5. Is the manuscript presented in an intelligible fashion and written in standard English?

Reviewer #1: Yes

Reviewer #2: Yes

6. Review Comments to the Author

Reviewer #1: all comments have been addressed. the authors have addressed the comments from the reviewers and this is now ready for acceptance.

Reviewer #2: All reviewer comments have be addressed in an acceptable manner and is reported in the manuscript. I recommend this manuscript be approved

7. PLOS authors have the option to publish the peer review history of their article (what does this mean?). If published, this will include your full peer review and any attached files.

Reviewer #1: **Yes: **HARPREET SINGH GREWAL

Reviewer #2: **Yes: **Venkata Buddhavarapu

---

## [Editor Report · Acceptance letter]

4 Apr 2024

PONE-D-23-36180R1 

PLOS ONE

Dear Dr. Rye, 

I'm pleased to inform you that your manuscript has been deemed suitable for publication in PLOS ONE. Congratulations! Your manuscript is now being handed over to our production team.

Kind regards, 

on behalf of

Dr. Vikramaditya Samala Venkata 

Academic Editor

PLOS ONE